# Intracytoplasmic Sperm Injection in Cattle

**DOI:** 10.3390/genes12020198

**Published:** 2021-01-29

**Authors:** Veena Unnikrishnan, John Kastelic, Jacob Thundathil

**Affiliations:** Department of Production Animal Health, Faculty of Veterinary Medicine, University of Calgary, Calgary, AB T2N4N1, Canada; veena.unnikrishnan@ucalgary.ca (V.U.); jpkastel@ucalgary.ca (J.K.)

**Keywords:** bovine, ICSI, sperm oocyte activation factor, phospholipase C zeta

## Abstract

Intracytoplasmic sperm injection (ICSI) involves the microinjection of sperm into a matured oocyte. Although this reproductive technology is successfully used in humans and many animal species, the efficiency of this procedure is low in the bovine species mainly due to failed oocyte activation following sperm microinjection. This review discusses various reasons for the low efficiency of ICSI in cattle, potential solutions, and future directions for research in this area, emphasizing the contributions of testis-specific isoforms of Na/K-ATPase (ATP1A4) and phospholipase C zeta (PLC ζ). Improving the efficiency of bovine ICSI would benefit the cattle breeding industries by effectively utilizing semen from elite sires at their earliest possible age.

## 1. Introduction

Sustainable Development Goals of the United Nations mandate a substantial increase in global food production in the near future [1]. Canadian animal production industries are at the forefront of improving animal productivity and contribute several billion dollars annually to our national economy. They rely on a variety of reproductive technologies such as artificial insemination (AI) [2] and embryo production [3] for genetic improvement and propagation of superior germ plasm globally. Since even a modest increase in the reproductive rate in cattle enhances productivity [4], improving efficiency of reproductive technologies will have immediate and substantial benefits globally. Production of frozen semen from elite bulls (progeny testing, based on their daughters’ milk production) and its use for AI have been practiced globally for dissemination of superior genetics and enhancing animal productivity. However, this industry has been recently revolutionized by genomic tools [5], facilitating early selection of superior bulls and widespread dissemination of their genetics. Reproductive technologies have substantially improved efficiency of genetic selection of bulls by reducing the generation interval from 5 years (progeny testing) to ~1.5 years [6]. Currently, industries are under pressure to market semen from bulls at their earliest possible age and AI centres are populated with younger bulls [7,8]. Despite the obvious benefits, this brings several challenges. Semen is now often collected from peri-pubertal bulls, yielding fewer sperm [9] of suboptimal quality [10], limiting its use for frozen semen production and AI. In response, in vitro production of embryos by in vitro fertilization (IVF) and embryo culture (IVC) or intracytoplasmic sperm injection (ICSI) are being used. Although IVF requires a minimum number of morphologically normal and functionally competent sperm for successful fertilization, ICSI with spermatogenic cells or sperm with suboptimal quality [11] was successful in several species [12,13,14,15]. Therefore, ICSI could be a viable option for the efficient use of semen of suboptimal quality (collected from young bulls) or sex-sorted. Although gametes contribute to genetic variation, selection of gametes based on genetic diversity is now possible using genomic selection [16]. Changing genetic selection from a superior animal for breeding to a superior gamete within an individual is not far from reality, as software that can identify gametic variance is already available [16]. Advancements in gametic selection would be an impetus for assisted reproductive techniques like ICSI for the efficient use of selected gametes for embryo production. Although this technique was developed for several species, i.e., rodents [17,18]; humans [19]; horses [20]; and swine [21], its success is not optimal in cattle, for various reasons. This review is focused on ICSI in cattle, with an emphasis on its applications, efficiency, reasons for ICSI failure, recent development, and future directions of research. The emphasis will be on sperm treatment and its contributions to the success of ICSI in cattle. ICSI is an in vitro technique in which a single sperm is microinjected into the cytoplasm of a matured oocyte. This technique is used to overcome infertility in humans and domestic animals. However, the technique was first performed using hamster sperm and oocyte, which produced a male pronucleus [14]. Thereafter, ICSI replaced conventional failed in vitro fertilisation in humans, where the sperm from oligospermic and asthenozoospermic patients were unable to fertilize oocytes. In addition, ICSI has been used in livestock and wild animals for improving livestock productivity, biodiversity conservation, transgenic animal production, e.g., pig [22] and for fertilization problems in IVF systems, e.g., horses [23,24], plus fundamental research in reproductive biology.

The first ICSI was performed with an in vitro-matured bovine oocyte by injection of sperm, resulting in embryo development up to a blastocyst and its transfer into a surrogate, with birth of viable offspring [25]. Thereafter, there were reports of successful birth of live calves [26,27]. Furthermore, ICSI-derived fully expanded blastocysts have survivability and quality similar to IVF-derived blastocysts after slow freezing [28] or vitrification [29]. As cryopreservation has become an integral part of assisted reproductive technology [28], development of better freezing protocols for cryopreservation of in vitro-produced embryos would also help to preserve ICSI-derived embryos, including genetic conservation of wild bovine species.

## 2. Applications of Bovine ICSI

Earlier, ICSI was used as a last resort when IVF failed. Viable embryos were produced from various types of spermatogonic cells, e.g., spermatids [30] and sperm obtained from in vitro culture of secondary spermatocytes [11]. Microinjection of bovine heat-dried [31] or freeze-dried [32] sperm yielded blastocysts. ICSI would be beneficial in preservation and conservation of endangered bovine species using lyophilised [32] or heat-dried sperm [31]. Frozen-thawed bovine oocytes are suitable for ICSI, as this improves pronuclear formation [33] and cleavage rate [34] compared to IVF with frozen-thawed bovine oocytes. Hence, in vitro production of bovine embryos with gametes of variable quality (blastocyst rate formation in good vs. poor quality oocytes was 23.3% and 11.1% respectively) can be effectively achieved with ICSI [35].

Sex-sorted sperm have been used to produce female dairy and male beef calves, using in vitro production of embryos [36]; this has substantially advanced cattle productivity, including propagation of genetics from superior cattle that are culled due to injuries [37]. Availability of frozen sex-sorted bovine sperm increases the use of IVF in cattle breeding programs [38]. However, embryos resulting from in vitro fertilization using sex-sorted sperm had poor developmental competence, and the resulting embryos had poor calving rates [36]. Jo et al. (2014) reported that ICSI of sex-sorted sperm (24.7%) yielded more embryos than IVF (2.7%) of sex-sorted sperm [39], encouraging its use to produce sex-specific embryos [40].

ICSI has been effectively used for production of transgenic embryos [41,42,43,44]. Sperm-mediated gene transfer uses sperm to transport exogenous DNA into the oocyte during fertilization, resulting in transgenic embryos [42]. In farm animals such as sheep, goats and cattle, transgenesis has been used to generate animals that express recombinant protein in milk, or to produce porcine organs for human xenotransplantation [43]. Intracytoplasmic sperm injection mediated gene transfer (ICSI-MGT) has benefits over pronuclear microinjection as it eliminates low transgenic efficiencies and imprinting defaults inherent in somatic cell nuclear transfer (SCNT) [41]. Bovine blastocyst production with ICSI-MGT was comparable to or better than SCNT or pronuclear microinjection [41]. High blastocyst production was achieved in farm animal ICSI-MGT by chemical activation of oocytes using ionomycin and 6-dimethylaminopurine (DMAP) [41,43]. In addition, physical or chemical damage to the sperm membrane before microinjection improved ICSI-MGT [42].

Bovine ICSI has contributed to the assessment of oocyte activation [45] and centrosome function [45,46,47,48,49] of human sperm. Bovine ICSI has also been used as a heterologous assay system [47,50,51] for evaluating fertilising ability of human sperm [45] and human centrosome function [46,52]. These assays have led to identification of the role of male pronuclei in synchronising development of female pronuclei [50].

## 3. Potential Reasons for Failure of ICSI in Cattle

The success rate of ICSI in cattle (14%) is low compared to other domestic species (horse: 21%, goat: 28% and pig: 18%) [53]. However, since ICSI has tremendous potential for augmenting genetic selection and animal productivity, research is ongoing to improve its efficiency. Major reasons for the failure of ICSI in cattle were regarded as the inability of sperm to undergo nuclear decondensation and pronuclei formation [54], improper functions of microtubule organising centre [55], and failure to elicit calcium oscillations [56] required for oocyte activation. In Vitro-matured bovine oocytes are incapable of processing sperm with an intact acrosome or sperm that has not undergone capacitation [54]. It has been reported that the acrosomal enzymes deform and lyse oocytes [57]. Furthermore, when sperm from various species (hamster, cattle, swine, human and mouse) were microinjected to mouse oocytes, the order in which they cytolyse the oocyte was correlated with acrosome volume. Injecting trypsin and hyaluronidase (which mimicked action of acrosome-intact sperm) into a normal, fertilized mouse oocyte disturbed pre- and post-implantation development [57]. However, removal of sperm membranes may improve male pronuclei formation [58] and make the sperm-derived oocyte activating factor (PLC zeta) more readily available to the oocyte cytoplasm. Compromised release or activation of sperm factor may cause failure of calcium oscillation [56].

During fertilization, calcium oscillation precedes oocyte activation. Fusion of sperm and oocyte releases sperm oocyte activation factor (PLC zeta) which is involved in the hydrolysis of phosphatidyl inositol biphosphate (PIP2), generating inositol triphosphate (IP3) and diacetyl glycerol (DAG). IP3 binds to receptors in the intracellular calcium reserves (e.g., endoplasmic reticulum), releasing calcium [59,60,61]. Oocytes have a second messenger-controlled activation model, with calcium and IP3 as second messengers. Increased intracellular cytosolic calcium concentrations induce calcium oscillations by activating calcium-induced calcium release (CICR), whereby calcium induces its own release from internal reservoirs [62]. Furthermore, calcium oscillations are maintained by calcium transients, which, depending on the time taken by the calcium stores to replenish calcium to facilitate the next spontaneous discharge [62]. The IP3-mediated increase in intracellular calcium is a feedback to inhibit further calcium release by inactivating calcium channel receptors, thereby forcing calcium back to internal reservoirs. This decrease in calcium removes the feedback inhibition on IP3-sensitive calcium channels and calcium oscillations are maintained by the periodic release of calcium from an IP3-sensitive calcium pool [62]. Thus, calcium oscillation promotes oocyte activation, manifested by the resumption of meiosis and formation of male and female pronuclei.

Changes during bovine capacitation and acrosome reaction during in vivo fertilization might favour release or activation of sperm factor. The sperm oocyte activation factor is apparently located at the perinuclear theca [63]. Persistence of the subacrosomal region of perinuclear theca (SAR-PT) on the apex of the male pronucleus disrupts the S-phase, not only in the male, but also in the female pronuclei [64,65]. In a primate study, complete solubilisation of post-acrosomal sheath perinuclear theca (PAS-PT) seemed to occur in parallel with oocyte activation [66]. Therefore, it is likely that greater rigidity of the perinuclear theca of bovine sperm [67] contributes to its difficulty in solubilisation of the perinuclear theca contents [66], and sperm nuclear stability [68] to its difficulty in sperm decondensation. Bull sperm microinjected into oocyte removed its plasma membrane after 20 h, but there was no sign of perinuclear theca removal, likely affecting the ability of oocyte factors to access sperm DNA and also preventing sperm decondensation [67]. The latter involves replacement of sperm protamines by oocyte-derived histones and is a pre-requisite for male pronuclear formation [69].

Condensation of the sperm nucleus is the result of protamine binding to the DNA [70]. The protamine is rich in cysteine and arginine. It binds to DNA mainly by electrostatic interaction of its arginines. The aggregation is further stabilized by inter- and intra-disulphide bonds [71]. In mammals, two distinct protamines are present, protamine 1 and protamine 2 [72]. Bull sperm has only protamine 1 and the nuclear chromatin is very stable with maximum number of disulphide cross links [70]. Bovine protamine 1 has central arginine-rich DNA binding domain and cysteine-rich domain at both ends [73]. Each cysteine sulfhydryl group is oxidised to intra- and intermolecular disulphide bridges [74]. Affinity to DNA is greater for protamine 1 versus protamine 2 [72]. The latter (present in human and mouse), has lower cysteine content than the former, so protamine 2 is expected to have lower disulphide bridges [75]. Therefore, sperm with a higher proportion of protamine 2 decondense quickly [54], accounting for lower stability of human and mouse sperm nuclei (contain protamine 2) compared to bull sperm.

Furthermore, to accurately nucleate and organise the sperm aster, the ooplasmic pericentriolar materials should be properly blended with sperm centrosomes [76]. Microtubule organising centres (MTOC) of sperm-microinjected oocytes (aster formation rate and fluorescent intensity of microtubule network) were not as functional as those of IVF oocytes [55]. Similarly, inadequate oocyte activation and male pronuclei formation may be due to compromised in vitro oocyte maturation, decreasing inositol-1,4,5-triphosphate (IP3) and glutathione reserves in cumulus cells [77]. A recent report using transcriptional regionalisation of developmental genes within M-II bovine oocytes and a preferential sperm entry point during IVF [78] implied oocytes may be polar. Consequently, the sperm entry point during ICSI may be important in embryo development [78]. Developing culture conditions [79] to improve ooplasmic environment [54] and mimicking molecular changes in sperm associated with physiological capacitation and acrosome reaction [54] through appropriate sperm pre-treatments, may increase efficiency of ICSI in cattle.

## 4. Pre-Treatment of Sperm for Improving the Success of Bovine ICSI

Effects of various pre-treatments on sperm were provided in Table 1. Mechanical pre-treatment of sperm by immobilisation of sperm by tail scoring improved male pronucleus formation [80,81] as localized disruption of sperm plasma membrane is expected to promote its disintegration within the oocyte [82]. The sperm plasma membrane is unable to heal, compared to other cells, due to minimal cytoplasm [83]. Through the disrupted membrane, extracellular sodium and calcium ions move inside the sperm, activating endogenous nucleases that cleave DNA. Consequently, it is important to minimize the interval between sperm immobilisation (disruption of sperm plasma membrane) and sperm injection into an oocyte [83]. For species with a stable plasma membrane (e.g., cattle), it is optimal to disrupt this membrane prior to ICSI [84].

Chemical pre-treatment of sperm improved ICSI outcome in cattle [80] by increasing nuclear decondensation and pronuclei formation. As bovine ICSI bypasses critical fertilisation events, e.g., capacitation, acrosome reaction and penetration of zona pellucida [77], there is a need to artificially subject sperm to these changes. Dithiothreitol (DTT) pre-treatment of sperm before bovine ICSI improved cleavage rate [85] and blastocyst development in OPU-ICSI when compared to OPU-IVF [86]. Treatment of oocytes and sperm with DTT without any oocyte activation agent resulted in the birth of a healthy calf [87]. Conversely, there are reports that sperm pre-treatment with DTT [88,89] along with oocyte activation with DMAP improved the efficiency of bovine ICSI. DTT reduces the disulphide bond necessary for sperm nuclear decondensation [90] and destabilizes the nuclear packaging of sperm head. DTT also nucleates the microtubules by organising Ƴ-tubulin in the sperm centrosome. The Ƴ-tubulin would access the microtubule components present in the ooplasm by the conformational change induced by reducing the disulphide bonds in sperm centrosome [76]. DTT is a thiol compound with two thiol groups. The reactivity of dithiol depends on its pKa value. When the pKa of a DTT is approximately equal to the pH of the media in which it is dissolved, the thiol-disufide interchange is maximal. The pKa of DTT is 9.0, greater than fertilization medium (7.8), so its active form (reduced DTT) is easily converted to inactive form (oxidised DTT) [74]. Recently a dithiol group, dithiolbutylamine (DTBA) increased the efficacy of blastocyst production in bovine ICSI [74]. DTBA is more efficient than DTT in preventing re-oxidation of sperm and can promote pronuclei formation, as it can be retained longer in sperm, due to its pKa of 8.2. Moreover, longer incubation of sperm with DTT increases DNA fragmentation and significantly reduces sperm viability compared to DTBA [74]. In addition, DTT predisposes sperm to chromosomal abnormalities [91]. Pre-treatment of sperm with glutathione (GSH) [92], plus oocyte activation by ionomycin in combination with ethanol, improved the efficiency of bovine ICSI. Hence, the detrimental chemical thiol compound pre-treatment (e.g., DTT) can be replaced by glutathione (GSH) [91,92], a major non-protein thiol compound naturally present in mammalian cells. GSH is involved in many cellular functions including disulphide bond reduction, protection against oxidative stress, etc. Although disulphide bond reduction is faster with DTT than GSH, the latter causes higher disulphide bond reduction rate (Figure 1). In ICSI, GSH can function with fewer side effects than DTT, reducing disulphide bonds and promoting sperm chromatin decondensation in vitro [91].

Ejaculated sperm must reside in the female reproductive tract for a species-dependent interval for its final maturation through a series of biochemical changes, (capacitation) enabling fertilizing ability. Pre-treatment of bovine sperm before ICSI with a capacitating agent, e.g., MβCD (methyl-β-cyclodextrin) [93], heparin [81,94] and heparin with glutathione (GSH) significantly improves fertilization and blastocyst formation rates [95,96]. Heparin + GSH improved the outcome of ICSI from sex-sorted sperm [96]. Heparin is a polyanionic glycosaminoglycan (GAG); these are compounds present in both male and female reproductive tract secretions and with important roles in fertilization [97]. Also, heparin and heparin-like GAGs in the oviduct contribute to capacitation of bovine sperm [98,99]. There are receptors for heparin present on sperm plasma membranes. It has been proposed that destabilisation of sperm plasma membrane occurs when heparin bind to its receptors, enabling incorporation of other molecules (e.g., GSH) into the sperm nucleus [75,100]. Heparin can decondense human sperm [101]. It has a strong affinity for protamine, forming a highly insoluble complex [75]. Sperm decondensation is dependent on the sulphation characteristics of heparin rather than a polyanionic molecule competing with DNA for positively charged protamines [101]. The desulphation effects of heparin affect the net charge, resulting in electrostatic interactions between charged groups and inducing conformational changes [101]. Furthermore heparin or heparin sulphate (a heparin analogue) were implicated in removal of sperm protamines [102] and heparin sulphate has been identified in oocytes, implying it may be involved in sperm decondensation, acting as a protamine acceptor [103]. In bull sperm, nuclear decondensation is induced by heparin [70] and heparin-GSH [97]. GSH treatment enhanced mitochondrial function in the sperm middle piece and significantly reduced the number of disulphide bonds in the sperm head [91]. The sperm plasma membrane may act as a barrier against sperm decondensation, but capacitated sperm with altered sperm plasma membranes were able to undergo nuclear decondensation when heparin and GSH used as decondensing agents [102]. Furthermore, addition of GSH [79] or heparin [94] to the ICSI culture medium improved embryo development, emphasizing the importance of pre-ICSI capacitation.

Pre-treatment of bovine sperm with detergents e.g., lysolecithin (LL) or Triton X-100 (TX) [104] along with glutathione before ICSI, induces plasma membrane disruption and promotes nuclear decondensation [104]. The former improved the rate of embryonic development, without affecting embryo quality [104]. These detergents act as membrane destabilising agents, promoting release of acrosomal content from the sperm head [104]. Recently, bovine ICSI outcomes were improved by using cysteamine (Cys) during in vitro maturation (IVM) of oocytes and pre-incubating sperm with mature COCs (cumulus oocyte complexes) before ICSI [105].

**Table 1 genes-12-00198-t001:** Effect of various pre-treatments on bovine sperm.

Pre-Treatment	Effect on Sperm	Reference
Mechanical pre-treatment		
Tail scoring	Removal of sperm membrane	[80]
Piezo pulses		[81]
Chemical pre-treatment		
DTT	Reduction of disulphide bond (sperm head decondensation) and involved in microtuble organisation	[85,86,87,88,89]
NaOH + DTT	Sperm decondensation and DNA fragmentation	[106]
DTBA	Disulphide bond reduction	[74]
LL + TX-100	Membrane destabilization	[104]
LL + TX + glutathione	Membrane destabilization and disulphide bond reduction	[107]
Pre-treatment with capacitating agents		
MβCD	Capacitation-associated changes	[93]
Heparin	Capacitation-associated changes and Sperm decondensation	[80,94]
Heparin + Glutathione	Capacitation-associated changes and disulphide bond reduction Sperm decondensation. Enhanced mitochondrial function	[95,96]
Heparin + Caffeine	Capacitation associated changes and acrosome reaction, Sperm decondensation	[80,108]
Glutathione	Disulphide bond reduction (sperm head decondensation). Enhanced Mitochondrial function	[91,92]
Cumulus oocyte complexes (COCs)	Acrosome reaction of sperm	[105]

## 5. Artificial Activation of Oocyte for Improving the Success of Bovine ICSI

Chemical or mechanical activation of the oocyte after ICSI has been commonly done in cattle. In that regard, use of electric stimulation [109], mechanical pre-treatment [80], or chemicals such as ethanol [110], ionomycin, anisomycin (ANY), cycloheximide (CHX), DMAP, dehydroleucodine (DhL) [111] independently or in various combinations, have been reported. Oocyte activation with ethanol alone or in combination with cycloheximide [112,113] or ionomycin [92,114] following ICSI resulted in development of blastocysts. Piezo-driven ICSI more efficient than conventional ICSI [81], and resulted in production of bovine offspring [115,116] when combined with ethanol oocyte activation. Viable calves were efficiently produced by post-ICSI oocyte activation with ethanol, compared to activation with ionomycin alone or a combination of ionomycin + DMAP [117]. Ethanol activates oocytes by increasing concentrations of intracellular free calcium [118]. The stimulus of sperm is sufficient to lower the maturation promoting factor (MPF) activity after ICSI and ethanol maintains the low MPF activity until the start of the next cell cycle [110]. MPF is a non-species specific ubiquitous cytoplasmic heterodimer protein whose activity is sensitive to calcium; its elevated activity is required for the metaphase II arrest of meiosis in oocytes [119]. MPF consist of cdc2 kinase which is associated with cyclin B [120,121]; activation of this kinase is dependent on its phosphorylation state [122]. Activity of a c-mos protooncogene product called cytostatic factor [123,124] stabilises MPF and promotes the arrest of oocyte at metaphase II [119]. The intracellular calcium surge induced by ethanol inactivates the cytostatic factor. Alternatively, cycloheximide inactivates MPF by preventing the synthesis of nascent proteins and degradation of cyclin B. Therefore, a combined synergistic treatment of ethanol and cycloheximide forces the oocyte out of metaphase stage of cell cycle and results in its activation [113].

Ionomycin in combination with DMAP [125,126], cycloheximide [28,83] or roscovitine [127] have been used in bovine oocyte activation protocols [128]. Intracytoplasmic injection of round sperm resulted in efficient production of developmentally competent embryos with repeated ionomycin activation, followed by cycloheximide treatment [28]. The interval between the addition of ionomycin and DMAP has a crucial role in bovine ICSI [125]. However, delaying the addition of DMAP resulted in production of activated oocytes with reduced chromosomal abnormalities [126]. DMAP inhibits protein phosphorylation after oocyte activation. Also, DMAP accelerates post-fertilisation events by inhibiting DMAP-sensitive kinases, implicated in the formation of the interphase network of microtubules, remodeling of sperm chromatin and pronucleus formation [129]. Among the MPF inhibitors used for oocyte activation, roscovitine is one of the most effective, with fewer detrimental effects [130]. Anisomycin for oocyte activation has resulted in superior developmental rates of resulting embryos, compared to cycloheximide and DMAP [131,132]. Anisomycin is a protein synthesis inhibitor that acts specifically in the translational stage [133]. Oocyte activation can also be induced without any artificial oocyte activation agent. For example, injection of PLCZ1 (sperm oocyte activation factor) cRNA resulted in calcium oscillatory pattern and embryos with low levels of aneuploidy [134]. Injection of bovine sperm cytosolic extracts (contain sperm oocyte activation factor) activated bovine oocytes and resulted in second polar body extrusion [135]. These studies suggested the possibility of replacement of chemical pre-treatment of sperm and chemical oocyte activation agents with capacitating agents and sperm oocyte activation factor to produce embryos in vitro by ICSI.

## 6. Sperm Oocyte Activation Factors (SOAFs)

According to the sperm factor hypothesis, a soluble sperm factor is released into the oocyte and triggers oocyte activation [136,137,138]. Advances in calcium imaging and clinical experiments involving ICSI (intracytoplasmic sperm injection) have provided substantial evidence for this hypothesis, leading to dismissal of the receptor-based mechanism of oocyte activation [139,140]. The search for possible sperm oocyte activation factors (SOAF), resulted in several candidate proteins, e.g., oscillin, glucosamine-6-phosphate isomerase (GPI) and citrate synthase [136,141]. Despite experiments supporting calcium oscillation activity in mammalian oocytes, there is no convincing evidence for describing its role in mammalian fertilisation [142].

Phospholipase C (PLC) isoforms catalyse hydrolysis of PIP2 (phosphatidyl inositol biphosphate) to inositol triphosphate (IP3) and diacyl glycerol (DAG). Thereafter, IP3 releases calcium via a receptor localised on the surface of the endoplasmic reticulum and DAG and calcium together activate the protein kinase C (PKC) pathway, resulting in cellular responses [137,143,144]. Moreover, previous studies characterized and identified a sperm-specific PLC isoform, PLC zeta (PLC ζ) as a candidate for the oocyte activation factor [60,140,142,145]. Furthermore, a post acrosomal WW-domain binding protein (PAWP) was a candidate for SOAF [146]. Microinjection of the PAWP cRNA or recombinant PAWP into porcine, bovine, *Xenopus*, mouse or human oocytes resulted in calcium oscillations, similar to those with ICSI and oocyte activation. Sperm inhibited with a competitive inhibitor for PAWP-derived PPGY peptide prevented the calcium oscillation [146,147]. PAWP is localised to the post acrosomal sheath-perinuclear theca of sperm head; it has an N-terminal with a sequence homology to WW-domain binding protein 2 and proline-rich C-terminal with a PPXY binding site and unknown repeating motif [146,147,148]. A hypothetical model on how this PAWP triggers calcium oscillation has been suggested [146,147]. It is thought to bind to oocyte-borne YAP protein and interact with the SH3 domain of PLC γ, activating a phosphatidyl inositol pathway [146,147]. Ever since the introduction of PAWP, it emerged to question the PLC ζ as SOAF. A study comparing the calcium oscillation in mouse oocyte by the microinjection of recombinant PAWP and PLC ζ reported that recombinant PLC ζ resulted in calcium oscillations similar to those in mammalian fertilization, whereas recombinant PAWP did not. Consequently, PAWP is not a convincing SOAF candidate [149]. Moreover, independent laboratories have validated the role of PLC ζ as a SOAF in a replicable and reliable manner [60,139,150,151,152,153].

## 7. PLC ζ as a Sperm-Specific Oocyte Activating Factor

In most mammalian species, an ovulated oocyte is arrested at Metaphase-II [154]. Oocyte activation normally occurs immediately following sperm penetration of the oocyte [155], triggered by a series of calcium waves [156], leading to resumption of meiosis, formation of male and female pronuclei, and fusion of these pronuclei leading to zygote formation. It has been established that a sperm-specific PLC ζ (located in the sperm head) is the sperm-derived oocyte activating factor [60,139,150,151,152,153]. The PLC ζ has been identified in several mammalian species (rats: [157]; pigs: [158]; cattle: [132]; monkeys and humans: [159]). Immunodepletion of PLC ζ from sperm protein extracts completely abolished calcium oscillation-inducing activity of sperm extracts [60]. There is mounting clinical evidence implicating involvement of abnormal, aberrant and mutant forms of PLC ζ resulted in failure of egg activation [150,151,153]. Microinjection of sperm heads lacking PLC ζ failed to activate oocytes, due to either no calcium oscillation or diminished calcium profiles in humans [150,151]. It has been proposed that PLC ζ elicits calcium oscillations in oocytes through a phosphoinositide signalling pathway, by hydrolysing membrane-bound phosphatidylinositol 4,5-bisphosphate (PIP2), resulting in the release of inositol triphosphate (IP3), which in turn binds to the IP3 receptors in intracellular calcium reserves, leading to intracellular calcium oscillations [59,60]. However, mechanisms of PLC activation eliciting calcium waves immediately following sperm penetration remain unknown.

## 8. A Hypothetical Model for PLC ζ Activation during Sperm Capacitation

Bull sperm can be capacitated in vitro by incubating with capacitating agents [heparin; ouabain, a cardiac glycoside; or a combination of cAMP and IBMX (a phosphodiesterase inhibitor) at 39 °C under 5% CO_2_ and high humidity]. The activity of PIP2-PLC is higher in capacitated versus uncapacitated sperm [160], suggesting a capacitation-associated increase in PLC ζ activity, with key roles in regulation of sperm capacitation and fertilization. However, the molecular basis of this PLC ζ activation and the role of specific capacitating agents in this process remains unknown.

The ubiquitous and testis-specific isoforms of Na/K-ATPase (ATP1A1 and ATP1A4, respectively), are present in bull sperm. Interaction of ouabain, a cardiac glycoside, with Na/K-ATPase regulates sperm capacitation [161]. ATP1A4 interacts with several proteins [162], including phospholipase C zeta (PLC) [163]. Significance of the interaction between ATP1A4 and PLC ζ during bull sperm capacitation is unknown. Capacitation includes tyrosine phosphorylation (Y-p) of sperm proteins [69,164], actin remodelling [165] and hyperactivated motility [166]. Incubation of bovine sperm with ouabain induced tyrosine phosphorylation and an acrosome reaction [167] via PKA, RTK and Src kinases [168] and ERK [169]. That PLC ζ exerts its effect immediately following sperm penetration of the oocyte implies it is already active at the time of sperm penetration. In LLC-PK1 cells, ATP1A1 tethers PLC- γ1 and IP3 receptors to form a Ca2^+^-regulatory complex and ouabain-induced phosphorylation of PLC- γ1 at Tyr (783) activated PLC- γ1 in a Src-dependent manner [170]. Ouabain-induced capacitation co-localized PLC ζ and ATP1A4 to the post-acrosomal region of sperm head [163] and confirmed their interaction, perhaps activating PLC ζ, generating IP3 and diacyl glycerol (DAG) in sperm. IP3 binds to IP3R, increasing intracellular Ca2^+^, whereas DAG-mediated activation of PKC converts globular actin (G-actin) to filamentous-actin (F-actin), essential for capacitation Figure 2; [171]. Therefore, the role of ATP1A4-induced activation of PLC ζ in filamentous-actin formation during capacitation and triggering calcium oscillations (through PLC ζ-mediated cleavage of PIP2 and generation of IP3) during sperm-oocyte fusion warrant further research (Figure 3).

In summary ICSI is a very reliable and efficient reproductive technique. Improved fertilisation from sex-sorted sperm and gametes with variable quality is the advantage of ICSI over other reproductive techniques. Additionally, its use in gene transfer makes it more valuable. ICSI is not successful in cattle due to difficulties in sperm nuclear decondensation, functioning of microtubule organising centre, and oocyte activation. Furthermore, anatomical peculiarities of bovine sperm also contribute. However, various sperm pre-treatment and oocyte activating agents have improved bovine ICSI. Sperm for ICSI is not undergoing capacitation, an essential event in natural fertilisation. Sperm pre-treatment using capacitating agents have improved efficiency of bovine ICSI. That the major reason for the failure of bovine ICSI is mainly due to sperm, focusing research in this direction is warranted.

## Figures and Tables

**Figure 1 genes-12-00198-f001:**
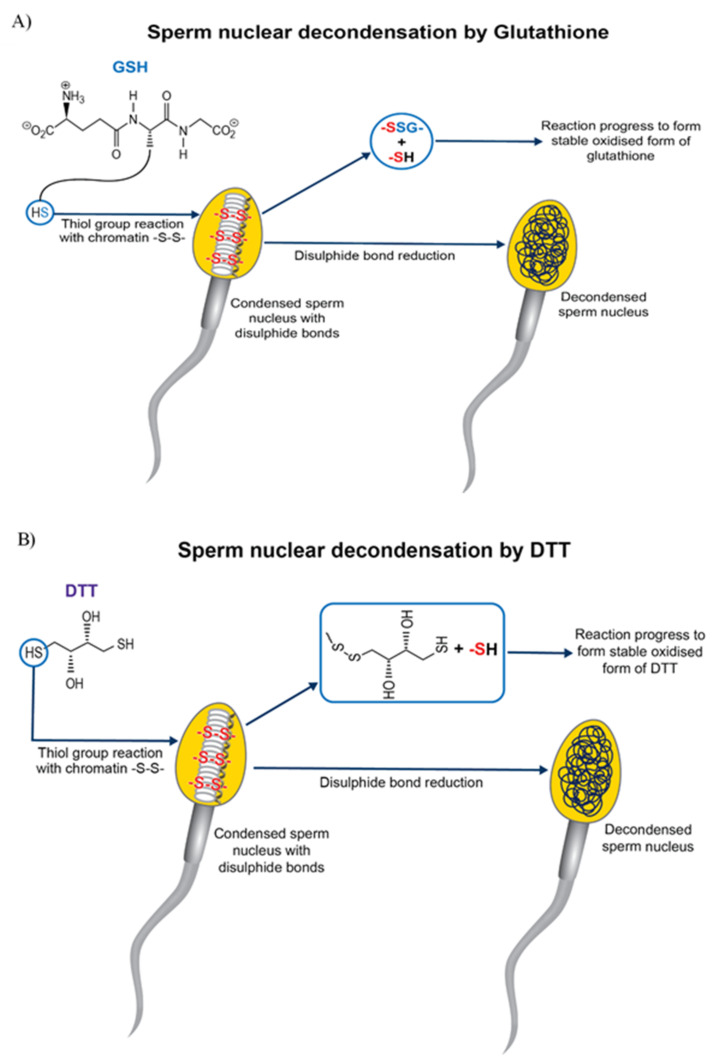
Sperm nuclear decondensation by disulphide bond reduction by thiol reagents. (**A**) Glutathione (GSH) reduces the disulphide bond in a condensed sperm nucleus, forming a mixed disulphide compound (-GSS-), along with a sulfhydryl compound (-SH). The reaction further progresses in the presence of thiol transferases enzymes, forming more stable oxidised GSH, using another molecule of GSH. Reduction of the disulphide bond results in decondensation of the sperm nucleus. (**B**) Dithiothreitol (DTT), a dithiol compound, reduces disulphide bonds in the condensed sperm nucleus, resulting in its decondensation. The reaction results in the formation of a sulfhydryl compound (-SH) and mixed disulphide compound (involving sulphur from DTT and sulphur from sperm nucleus), resulting in immediate reorganisation of its disulphide bond to form more stable oxidised DTT. This thiol-disulphide exchange is maximum when media pH equals pKa of DTT (9.0).

**Figure 2 genes-12-00198-f002:**
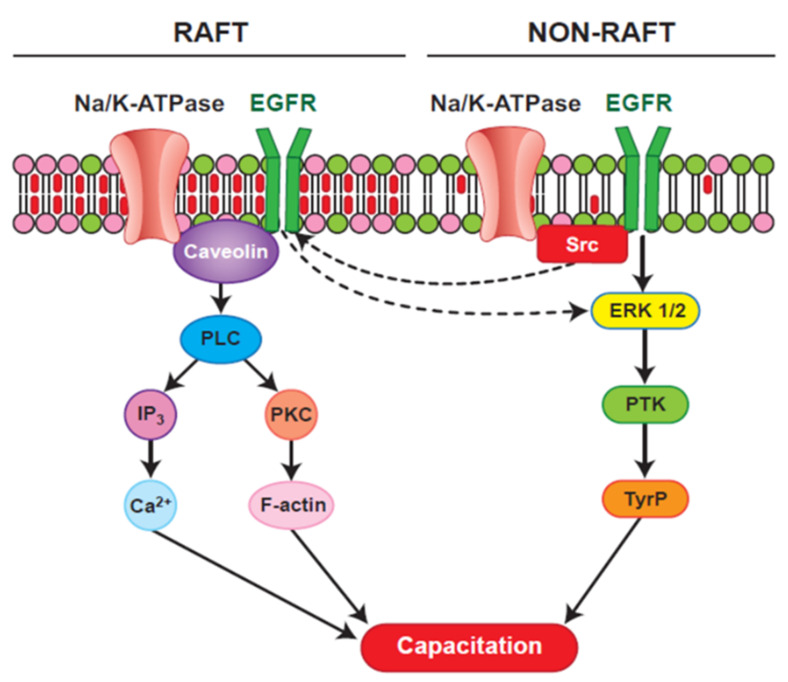
A hypothetical model for ATP1A4 (Na/K-ATPase α4)-mediated raft- and non-raft signaling pathways during bovine sperm capacitation [172]; reproduced with permission from John Wiley and Sons and Copyright Clearance Center]. Both raft and non-raft pools of ATP1A4 could activate downstream pathways during sperm capacitation. In the raft, ouabain signaling involves ATP1A4-caveolin-1-EGFR (epithelial growth factor receptor) complex which could bind and activate PLC (phospholipase C), thereby increasing hydrolysis of PIP2 (phosphatidylinositol biphosphate), generating IP3 (inositol triphosphate) and DAG (diacyl glycerol), which in turn activates PKC (phosphokinase C). IP3 binds to IP3R (inositol triphosphate receptors), increasing intracellular calcium, whereas PKC mediates polymerisation of G-actin to F-actin through other mediator proteins. Within non-raft, ATP1A4 signaling activates ERK1/2 (extracellular signal regulated protein kinase 1/2, a mitogen activated protein kinase) through activation of Src (Src kinase, a non- receptor tyrosine kinase), leading to PTK (protein tyrosine kinase) mediated tyrosine phosphorylation of proteins. Increase in F-actin, intracellular calcium and protein tyrosine phosphorylation contribute to capacitation-associated changes in sperm.

**Figure 3 genes-12-00198-f003:**
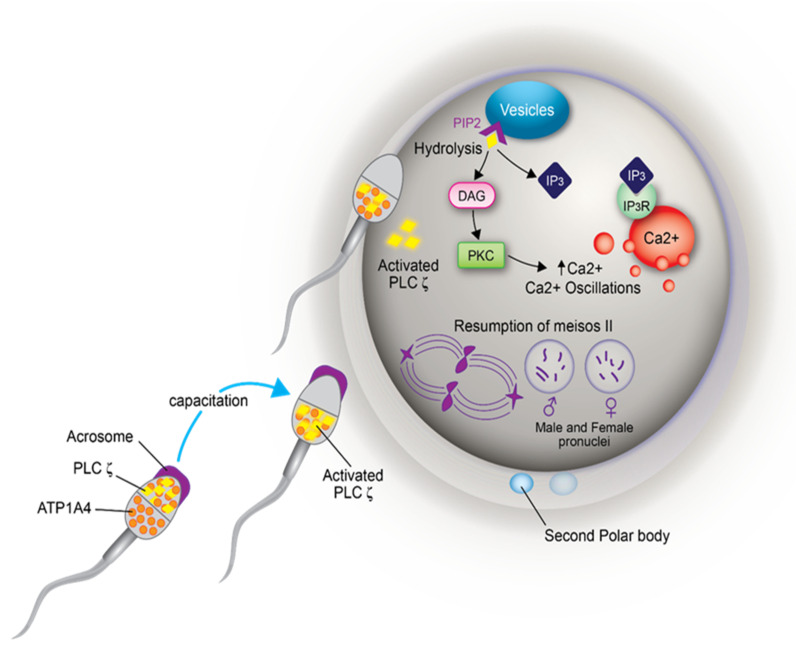
A hypothetical model depicting the involvement of ATP1A4 (Na/K-ATPase α4) mediated activation of PLC ζ (phospholipase C zeta) and the involvement of activated PLC ζ in oocyte activation. In fresh sperm, ATP1A4 is localized to the entire sperm head and PLC ζ to the acrosomal region. In capacitated sperm, both proteins are co-localized to the post-acrosomal region. The interaction of these two proteins during capacitation activates PLC ζ. Following sperm penetration, the equatorial segment of the sperm head fuses with the oolemma, resulting in the release of activated PLC ζ into the oocyte cytoplasm. This activated PLC ζ binds to PIP2 (phosphatidylinositol biphosphate) substrate, present in small vesicles inside the oocyte, resulting in its hydrolysis and formation of IP3 (inositol triphosphate) and DAG (diacyl glycerol). IP3 binds to its receptor in intracellular calcium reserves and releases calcium, thereby increasing intracellular calcium ion concentrations. In addition, DAG activates PKC (protein kinase C), which also increases calcium concentration resulting in calcium oscillation. This calcium oscillation results in resumption of meiosis of the metaphase II-arrested oocyte, expulsion of the second polar body and formation of a female pronucleus. Concurrently, sperm nuclear decondensation occurs, resulting in formation of the male pronucleus. Fusion of male and female pronuclei results in zygote formation.

## Data Availability

Not applicable.

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
