# Peer review of "Intracytoplasmic Sperm Injection in Cattle"

_genes, 2021, doi:10.3390/genes12020198_

Round 1
Reviewer 1 Report
The manuscript entitled “Intracytoplasmic Sperm Injection in Cattle” has been carefully evaluated. In this Review the authors summarize a lot of interesting and novel data regarding ICSI in the bovine species. The need of improving bovine ICSI in the introduction could be re-written to make it more convincing that ICSI in the bovine species really needs improvement because nowadays there are good results of traditional IVF in the bovine species, compared with the equine IVF. All in all, the manuscript is well written in a very clear scientific language, and the sections are presented in a good style. Nevertheless, there are some minor comments which need to be addressed. Please note my specific comments below.
-Please check the writing style of the Journal for Latin phrases, if in vitro should be written in italic letters, if so, please improve accordingly.
- After reference (10) the abbreviation IVF should appear behind “in vitro fertilization” and the abbreviation IVC should appear behind “in vitro culture”.
- Section 2: Could you please the success rate of produced embryos as mean embryo rate in % of ICSI produced blastocysts.
- Jo et al (39,40)… - Please provide in this sentence the embryo rates in % of traditional IVF embryos and ICSI to provide an immediate comparison for the reader without looking up the references.
-chemical activation of oocytes (41,43)…- Please provide briefly the compounds used for chem.act.
- Section 3: First sentence: …success rate in cattle is low…compared to other domestic species…- Please provide here the percentages, e.g.: success rate in cattle is low (XX%) compared to other domestic species (e.g. horse YY%, pig ZZ%, …)
- Section 4: Please introduce the abbreviation DMAP here, not in section 5.
- Please use the same style for IP3 and PIP2 in the whole manuscript since the indexes “3” and “2” are written differently.
-Fig.2: There are missing explanations of some abbreviations in the figure legend, e.g. ERK, Src, PTK, EGFR, please explain because for readers who are not familiar with the topic it could be hard to understand.
Author Response
- Point 1: Reviewer 1 has commented to re-write the introduction to make it clear how ICSI is better than IVF in bovine and also with comparison to other species like horses. However, focus of this introduction is potential applications for ICSI in utilizing suboptimal semen from peri-pubertal bulls at their earliest possible age for enhancing genetic selection. A comparison of IVF with ICSI was not the objective of this review.
- Point 2: Have checked the writing style for latin phrases and no need to give italic fonts to in vitro.
- Point 3: L38; abbreviations were provided for in vitro fertilization and embryo culture (IVF and IVC, respectively).
- Point 4: L81 to 83; The success rate of bovine ICSI using variable quality oocytes has written in percentage for better understanding.
- Point 5: L90 to 92; Percentage values of IVF and ICSI using sex-sorted sperm has indicated.
- Point 6: L101 to 103; The chemical activation agents used in ICSI-MGT has mentioned.
- Point 7: L112 to 113; The percentage success rate of ICSI in bovine comparing to other domestic animals have included.
- Point 8: Line 204 to 205; the abbreviation for 6-dimethylaminopurine (DMAP) has already mentioned in section 3 now.
- Point 9: We have changed all the PIP2 and IP3 to same style.
- Point 10: Line 419 to 425; We have given explanations for the abbreviations in Figure 2.

Reviewer 2 Report
The justification for this review is that Intracytoplasmic sperm injection (ICSI) does not work in the agriculturally relevant bovine system. The review, which is comprehensive and well written is directed toward the use of ICSI in cattle as a means to enhance breeding efficiency and ability to take advantage of superior genetic qualities when sperm collection is impossible or inadequate from immature animals. This is well presented and supported by a very comprehensive description of the cellular and molecular events involved in fertilization. There apparently is literature cited by the authors that capacitation of sperm is required, or even essential for bovine fertilization to be successful by ICSI.
Author Response
Reviewer 2 has not recommended any corrections.